# Factors Affecting Fatty Acid Composition of Holstein Cow’s Milk

**DOI:** 10.3390/ani13040574

**Published:** 2023-02-06

**Authors:** Ruth Rodríguez-Bermúdez, Ramiro Fouz, Margarita Rico, Fernando Camino, Taile Katiele Souza, Marta Miranda, Francisco Javier Diéguez

**Affiliations:** 1Departamento de Anatomía, Produción Animal e Ciencias Clínicas Veterinarias, Universidade de Santiago de Compostela (USC), 27004 Lugo, Spain; 2IES Valle del Oja, Santo Domingo de la Calzada, 26250 La Rioja, Spain; 3Departamento de Medicina Veterinaria, Universidade Federal Rural de Pernanbuco, Recife 52171, Brazil

**Keywords:** fatty acids concentration, dairy cattle, mixed effects ordinal regression, dairy farming, raw milk quality, methane emissions

## Abstract

**Simple Summary:**

Milk fatty acid composition has gained the interest of both manufacturers and consumers during recent years. The present paper aimed to perform an analysis of fatty acid composition in cow’s milk in relation to the type of ration, parity, lactation phase and season. Milk fatty acid profile varies significantly through the studied effects. According to different studies, fatty acid profile is associated with animal health, organoleptic properties of milk or even methane production, which highlights the importance of studying factors that affect its variation.

**Abstract:**

Milk fatty acid composition has gained the interest of both manufacturers and consumers during recent years. The present paper aimed to perform an analysis of C14:0, C16:0, C18:0, C18:1, saturated (SFA), monounsaturated (MUFA), polyunsaturated (PUFA) and short chain fatty acid (SCFA) concentration in cow’s milk in relation to the type of ration, parity, lactation phase and season. Cows’ milk from animals being fed total mixed rations, including corn silage, had higher C14:0, C16:0 and SFA concentrations than those being fed pasture-based rations but lower concentrations of C18:0 and PUFA. Comparing to 1st parity cows, 2nd and 3rd parity animals had higher SFA and SCFA concentrations in milk. With respect to spring, C14:0, C16:0 and SFA concentrations increased in summer, autumn and winter while MUFA, PUFA and SCFA concentrations decreased. Considering the lactation phase, C14:0, C16:0 and SFA concentrations decreased in fresh cows with ketosis comparing to healthy fresh cows and increased in peak, mid and late lactation. C18:0, C18:1 and MUFA follow the opposite trend. The milk fatty acid profile varies significantly through the studied effects. The fact that the fatty acid profile is associated with animal health, organoleptic properties of milk or even methane production highlights the importance of studying factors that affect its variation.

## 1. Introduction

The majority of milk lipids are in the form of triacylglycerols, which consist of a molecule of glycerol bound to three fatty acids (FA). During recent decades, milk FA composition has gained the interest of manufacturers and consumers as it influences nutritional, physical and flavor properties of dairy products [1].

Milk fat triacylglycerols are synthesized from more than 400 different FA, which makes milk fat the most complex of all-natural fats. Nevertheless, nearly all of these acids are present in trace quantities and only approximately 15 acids are at the 1% level or higher [2]. The most abundant are C16:0, C18:1, C14:0 and C18:0; ordered from largest to smallest. Thereby, based on previous results, mean FA in Holstein milk were 32.2%, 22.9%, 11.9% and 9.6% of the total milk FA, respectively [3]. Another previous study, considering confined and pasture cows separately, indicated figures of 31.7/31.2%, 23.3/23.1%, 15.4/13.4% and 9.4/10.2% for confined and pasture cows, respectively [4]. Thereby, percentages could oscillate according to different studies as, for example, the previous papers mentioned [3,4].

Milk FA composition is influenced by several factors, the main one being feeding, but also, individual variability, genetic parameters, breed, parity, milk and fat yield, stage of lactation or metabolic status of the animal [5].

The FA of bovine milk fat originate from two main sources: synthesis de novo in the mammary gland and the plasma lipids originating from the diet [6]. The FA from these two sources differ in their structure. The FA that are synthesized de novo are mainly short-chain and medium-chain length acids, from C4:0 to C16:0, while C18:0 and some C16:0 originate from the plasma lipids [7].

The main substrate for de novo synthesis of FA in the dairy cow is acetyl-CoA originating from acetate. Acetate provides the majority of the carbon and approximately one-half of the reducing equivalents needed for de novo lipogenesis [8]. Acetyl-CoA can also be metabolized to ketones and used to some extent as energy. However, increased ketogenesis and low tissue absorption may result in increased circulating ketone bodies and occasionally cause hyperketonemia [9]. Ketosis is related to the modification of the FA profile, showing a tendency to reduce SCFA and medium-chain (MCFA) and increase long-chain fatty acids (LCFA) [10]. C18:1 is one of the predominant FA in adipocytes and is released significantly during lipolysis in cases of negative energy balance (NEB). In these situations, a higher amount of LCFA is diverted into the bloodstream and incorporated into the milk fat, which in turn, contributes to inhibit the synthesis of SCFA and MCFA in the mammary gland.

On the other hand, milk FA are also related to methane production because of the common biochemical pathway between methane and FA in the rumen. In line, various dietary strategies were proposed to reduce production of methane by dairy cattle [11]. According to Chilliard et al. the most positive correlations between milk FA concentrations and methane output were observed for saturated FA from C6:0 to 16:0 and for C10:1 (r = 0.87 to 0.91) [12]. Among the four main FA, the one with the highest correlation is C16:0 [12].

Until now, the majority of scientific papers on this topic were based on experimental data with a limited number of samples. In this sense, it would be useful to perform analysis on field farm conditions using a larger amount of data.

The aim of the present paper was to perform an analysis of the main FA (C14:0, C16:0, C18:0 and C18:1), SFA, MUFA, PUFA and SCFA concentration in cow’s milk belonging to farms in the Dairy Herd Improvement Program (DHIP) and estimated CH_4_ emissions in relation to the type of ration, calving number, season and lactation phase.

## 2. Materials and Methods

### 2.1. Study Area and Herds Surveyed

The study was carried out from July 2018 to June 2019 in Galicia (NW Spain) in order to collect data for a complete year. Galicia is the major dairy cattle region in Spain with 56.4% of the country farms, accounting for 41.2% of the whole milk produced in the country (1.3% in the European Union) [13]. In Galicia, 35% of the herds are enrolled in the DHIP, which represents 72% of the milk produced in this region.

The data used in the study were obtained from 1557 cows (all of which are the Holstein breed) belonging to 25 dairy farms selected by simple random sampling among those included in the DHIP. The farms used for this study belonged to a major research project (FEADER 2016/59B). Table 1 shows characterization data from farms participating in the study. Records of these animals were obtained from test days run by the DHIP, during which the supervising technician measured the daily milk yield and collected a composite milk sample for fat percentage, protein percentage and FA composition (among other traits) following an alternative AM–PM monthly recording scheme throughout the lactation period. FA composition included the quantity of C14:0, C16:0, C18:0, C18:1, SFA, MUFA, PUFA and SCFA, MCFA, LCFA and total FA (in g/100 g of milk). From these data, the concentration of each of the main four FA were calculated, in addition to the concentration of SFA, MUFA, PUFA and SCFA (in g/100 g of total FA). Additionally, milk β-hydroxybutyrate (BHB) concentration was measured in all the animals from the 1st post-partum (PP) test day. In addition, for each cow, CH_4_ emissions were estimated based on FA composition, particularly on palmitic acid following the recommendations of Chilliard et al. [12]:Estimated CH_4_ production (g/day) = 16.8 acid palmitic (% total FA)—77

Finally, the study included data from 10,098 test-days (obtained from the 1557 cows). Each test day included: date, days in milk (DIM) and, as mentioned, daily milk yield, fat percentage, protein percentage, FA composition, estimated CH_4_ emissions and, in the case of a 1st PP test-day, milk BHB concentration. For analysis, DIM was divided into five categories: fresh cows without ketosis (DIM ≤ 35 and BHB concentration in the first PP test-day < 0.10 mM/L), fresh cows with ketosis (DIM ≤ 35 and BHB ≥ 0.10 mM/L), peak lactation (DIM > 35 to 90), mid lactation (DIM 91–210) and late lactation (DIM >210). De Roos et al. defined hyperketonemia as a milk BHB concentration of ≥ 0.10 mM/L [14]

Likewise, the calving number of each cow and the type of ration (according to type of forage used) in each farm were recorded. For the analysis, the calving number was divided into three categories: 1st, 2nd and ≥3rd. The type of ration was categorized as: pasture based, grass silage-based total mixed ration (TMR), corn silage-based TMR or grass/corn silage-based TMR. Farms that supplied both corn and grass silage showed grass/corn proportions from 60/40% to 40/60%. Samples were also categorized according to the season in which samples were collected as: spring, summer, autumn and winter in north hemisphere conditions.

### 2.2. DHIP Determinations

According to Decreto 120/2011 [15], LIGAL (Laboratorio Interprofesional Galego de Análise do Leite) is designated as the official laboratory for milk sample analysis of DHIP. Milk samples were collected by DHIP technicians and delivered to LIGAL for their analysis following official criteria UNE-EN ISO/IEC, with no 99/LE276 [16].

The milk used in DHIP determinations corresponded to samples from complete cow milking from all quarters. Samples were collected in plastic 50-mL containers to which a preservative, bronopol (2-brono-2-nitro-1,3- propanediol) was previously added. The fat percentage, protein percentage and FA and BHB determinations were performed using Fourier-transform infrared (FTIR) spectrometry (Milkoscan FT6000, Foss Analytics, Hilleroed, Denmark) [17].

### 2.3. Statistical Analysis

Statistical analyses were performed using R Statistical Software. The effect of the type of ration, calving number, lactation phase and season on mean concentration (g/100 g of total FA) of C14:0, C16:0, C18:0, C18:1, SFA, MUFA, PUFA and SCFA, aside from CH_4_ emissions, was examined by Kruskal–Wallis.

After that, in a multivariate approach, mixed-effect ordinal regression models were used to estimate the influence of these explanatory variables on the concentration of each FA (C14:0, C16:0, C18:0 and C18:1) or FA group (SFA, MUFA, PUFA and SCFA) (dependent variables. Thereby, 8 models were performed in total). For this purpose, the concentrations of each FA and group were divided into four categories according to the quartiles of its distribution. Table 2 shows the percentiles used as cut points helping to establish the four categories.

Therefore, the ordinal models report how each predictor variable uniquely affects the odds of (i) being in category 4 (>Q75), 3 (Q50–Q75) or 2(Q25–Q50) compared to category 1 (<Q25); (ii) being in category 4 or 3 compared to being in category 2 or 1; and (iii) being in category 4 compared to category 3, 2 or 1. Thereby, the regression coefficients indicate the likelihood of being in a higher category when the explanatory variables change.

Energy corrected daily milk yield (ECM) was also included in the regression models as control variable [18]. This trait was calculated as follows: ECM = (daily milk yield × (0383 × % fat + 0242 × % protein + 07832)/31138).(1)

All models were implemented at test-day-level, with herd and cow as random factors, to make adjustments within herd and animal cluster effects. For random factors, the cluster variance was provided in terms of an intraclass correlation.

For the models, when a variable changed the effect of the remaining coefficients by 10% or more, it was considered a confounder and stayed in the model, regardless of its level of significance.

R scripts for the regression models are provided as Appendix A.

## 3. Results

Mean milk FA contents by type of ration, calving number, season and lactation phase are provided in Table 3. In the univariate approach (Kruskal–Wallis), differences among FA contents and estimated CH_4_ emissions for the different explanatory variables were significant (*p* < 0.05).

The regression models indicated that cows’ milk from animals feeding on corn silage-based TMR and grass/corn silage based-TMR had significantly higher odds of being in the categories with greater C14:0, C16:0 and SFA concentrations than those feeding on pasture-based rations (reference category). Conversely, they had significantly lower odds of being in the categories with greater C18:0 and PUFA concentrations. With regard to C18:1, MUFA and SCFA, coefficients relative to type of ration were not statistically significant and were removed from the final models since this exclusion barely modifies the remaining coefficients (Table 4).

Comparing to 1st parity cows (reference), 2nd parity animals had significantly higher odds of having greater C16:0, SFA and SCFA concentrations and lower odds of being in the categories with higher C18:1, MUFA and PUFA concentrations in milk. Using the same reference category, 3rd parity cows had higher odds of being in the categories with higher C18:0, SFA and SCFA but lower odds of having higher C14:0, C18:1 and MUFA concentrations (Table 4).

With respect to spring, the odds of having higher C14:0, C16:0 and SFA concentrations significantly increased in summer, autumn and winter, while the odds of having higher MUFA, PUFA and SCFA concentration decreased. In the case of C18:0 and C18:1 (also with respect to spring) the odds of being in categories with higher concentrations decreased in autumn and winter (Table 4).

Considering the lactation phase, the odds of having greater C14:0, C16:0 and SFA concentrations significantly decreased in fresh cows with ketosis comparing to healthy fresh cows (reference category) and increased in peak, mid and late lactation. C18:0, C18:1 and MUFA follow the opposite trend. In the case of PUFA (using the same reference category) the odds of having higher concentrations decreased in fresh cows with ketosis, peak, mid and late lactation, and for SCFA, it decreased in fresh cows with ketosis and increased in peak and mid lactation (Table 4).

## 4. Discussion

Although the variations of certain FA (i.e., SFA, MUFA, PUFA and conjugated linoleic acid) with parity, stage of lactation and season have been widely studied [19], the knowledge of changes in bovine milk FA composition according to these variables in farming conditions is still limited and mainly based on experimental research conducted on limited data sets [20].

A limitation of the current study, which should not be underestimated, was the potential role that other factors may have played on FA concentrations. However, it should be noted the importance of analyzing a large number of samples originating in field farming conditions as well as type of forage classification applied. Additionally, the type of ration was divided into four main groups despite the existence of other minor differences in the rations between farms. The large sample size, allowing the inclusion of multivariate mixed models, which allows control of intra-herd cluster effects, would be the study’s main strength. Likewise, unlike many previous studies, ECM was included as a control variable.

As diet plays a major role in determining the FA composition of bovine milk, a larger number of studies compared to other factors have been reported [20]; however most of them make simple comparisons between pasture-based systems and zero-grazing systems. In our study, pasture-based rations showed higher PUFA concentrations than those containing corn silage. On the contrary, corn-silage-rich diets had higher SFA, as found by Ellis et al. [21]. As expected, milk concentrations of C14:0 and C16:0 were higher as corn silage was included, showing opposite patterns to C18:0. Similar results were found in previous studies, in which corn silage increased proportions of C14:0 and C16:0 and decreased the amount of C18:0 (and also C18:1) when compared to alfalfa silage, grass silage, and fresh pasture [22,23,24]. Moreover, another study showed that increasing the proportion of fresh pasture (mainly ryegrass) to replace corn silage increased unsaturated fatty acid (UFA) concentration at the expense of SFA [25]. Some authors have found greater differences than those observed in the present paper between cows consuming grazed grass compared to conserved forage diets, showing that milk from grazing cattle contained higher proportions of UFA, C18:0 and C18:1 acids and lower of SFA, C16:0 and C14:0 acids than cows fed with diets dominated by conserved forages [26,27,28]. In our study, such marked differences were not found in cow’s milk after consuming fresh or conserved forage, probably due to the fact that concentrate supplementation plays a more important role in Northern Spain than in other regions such as Ireland, Netherlands or New Zealand [28,29]. The results found in our study regarding the changes in FA composition caused by the inclusion of corn silage in the diet are probably due to the differences in starch composition (10–34.2% in dry matter) [26]. A high intake of starch is associated with a higher level of de novo synthesis, resulting in more saturated milk fat. In contrast, higher intakes of PUFA from pasture result in a decrease in the proportion of SFA, mainly C16:0, in milk [30]. In this sense, similar results to ours were found in a previous study developed in the North of Spain with respect to C16:0 concentration when feeding dairy cattle with corn silage, grass silage, a combination of both and other type of diets [31]. As CH_4_ production is estimated from FA composition, particularly palmitic acid, it is expected that farms feeding corn silage emit higher CH_4_ levels in our conditions.

With respect to parity, in our study, C18:1 and MUFA were higher in primiparous cows, while C16:0, SFA and SCFA tended to be higher in multiparous cows. These findings are in accordance with other studies, where 1st parity cows had relatively higher proportions of UFA and oleic acids and lower of SFA and C16:0 in milk fat compared with later parity cows, even though some of these studies showed that 3rd parity in cows presented with higher amounts of milk PUFA and MUFA than 2nd parity cows [5,20,32]. Some authors pointed out that the mammary gland of 1st parity cows was metabolically less active, having a lower expression of FA synthase, than in later parity cows [33]; this could explain the lower proportions of de novo FA [20]. Another explanation for these findings is that, under commercial conditions, multiparous cows are usually fed higher proportions of concentrate in the diet because of their higher productions, leading to higher amounts of SCFA [20].

Regarding season, our results showed that C14:0, C16:0 and SFA were higher in summer and autumn while C18:0, C18:1 were higher in spring and summer and MUFA and PUFA mainly in spring. It is important to mention that seasonal variation in the milk composition is particularly related to nutritional factors associated with changes in availability and quality of pasture through the year. In fact, fat yield is highest in summer and lowest in winter [34]. Previous studies did not show uniform results on this topic. Auldist et al. have found that MUFA were highest in winter and spring, SCFA were lowest in winter and PUFA were highest in spring [34]. Collomb et al. and Frelich et al. have found that summer milk had lower concentrations of SFA and higher contents of MUFA and PUFA, particularly C18:1, when compared to winter [35,36]. These findings are influenced by the ratio between roughage and concentrates [35]. Grass yield and voluntary intake of grass begins to decline toward the end of the summer into autumn, and farmers tend to supplement the diet to a larger extent than in spring and early summer. This might explain the increase in SFA [37]. Thereby, with respect to estimated CH_4_ production, which is always in correlation to palmitic acid concentration, emissions are predicted to be lower in spring. This fact is in accordance with the variation in pasture composition depending on season [38].

The effect of stage of lactation was studied more extensively than the role of parity. It has been known for years that the concentration of fat is highest in late lactation and lowest in early lactation, and FA profile changes during lactation [20,34,39]. The milk FA across lactation can be related to cow’s physiology and to energy balance status. In early lactation, cows are in NEB, causing the mobilization of adipose FA and the incorporation of these FA in milk [5,20,39].

Overall, milk from cows in early lactation had higher proportions of MUFA, particularly C18:1, than milk from medium or late lactation [34]. These results are in accordance with our study, where the stage of lactation has a great influence. C14:0, C16:0, SFA and SCFA tend to increase as lactation progresses. With respect to early lactation, they are usually higher in mid lactation. On the contrary, C18:0, C18:1, MUFA and PUFA decrease as lactation progresses, with the lowest concentrations usually in mid lactation. Bastin et al. and Hanus et al. indicated that cows in NEB mobilize higher amounts of C18:0 and C18:1 to milk because of lipolysis, thus, the proportion of de novo FA is lower in early lactation [32,40]. This tendency is more noticeable in high genetic merit cows that were found to be more sensitive to lipolysis [40]. 

For the particular case of cows suffering ketosis in PP (reflected by BHB ≥ 0.10 mM/L), our study showed lower proportions of C14:0, C16:0, SFA, PUFA and SCFA and higher proportions of C18:0, C18:1 and MUFA with respect to PP cows not suffering ketosis. A significant decrease in several de novo and medium chain FA was observed in milk from cows with ketosis, which is similar to our results [41]. The decrease in the synthesis of de novo FA in milk might suggest a less metabolically active mammary gland [42]. Similarly to our case, Chandler et al. reported increased concentrations of long chain as well as UFA in hyperketonemic cows [43]. The increment of long chain FA and UFA could be related to a greater acidogenic ruminal fermentation due to lower dry matter intake and higher passage rate [42]. In accordance to our study, several studies find an increase in MUFA and LCFA with ketosis due to the mobilization of FA during NEB [10,41,43].

As long as lactation progressed, the proportion of de novo (C4:0 to C14:0) and mixed origin (C16:0 and C16:1) FA increased, whereas preformed FA (>C17:0) decreased [20,32,44]. Selection for higher milk yield in early lactation results in a greater reduction in SCFA and SFA in milk [20]. Comparable to our results are those of Fearon et al., reporting that during late lactation, cows produced milk fat containing a significantly higher proportion of UFA than during mid-lactation [45].

## 5. Conclusions

After analyzing the impact of type of ration, calving number, season and lactation phase in milk FA composition, it was concluded that all four factors have a significant influence in FA profile. Cows’ milk of animals feeding on TMR containing corn silage yielded higher C14:0, C16:0 and SFA concentrations, and lower C18:0 and PUFA concentrations in milk than pasture based dairy cattle. With respect to calving number, 2nd parity cattle produced higher C16:0, SFA and SCFA than primiparous cows. Third parity and older cattle yielded higher concentrations of C18:0, SFA and SCFA compared to primiparous cows. Regarding season, the concentrations of C14:0, C16:0 and SFA were higher in cow’s milk produced in summer, autumn and winter with respect to spring, while the concentration of MUFA, PUFA and SCFA decreased. Finally, concerning the lactation phase, fresh cattle yielded lower C14:0, C16:0 and SFA concentration than dairy cattle in peak, middle or late lactation. C18:0, C18:1 and AGMI concentrations followed the opposite trend.

## Figures and Tables

**Table 1 animals-13-00574-t001:** Mean number of lactating cows, parity, replacement rate and ration composition of studied dairy farms (n = 25) in northwestern Spain according to type of forage used for feeding dairy cattle.

	PB Farms	GR-TMR	C-TMR	GR/C-TMR
Number of cows	57.8	52.3	137.2	93.6
Parity number	4.4	3.7	2.7	3.3
Replacement rate	24.8	28.3	37.7	32.2
Daily ration composition (kg/cow)				
Fresh forage	6.2	0	0	0
Grass silage	3.6	9.6	2.4	5.1
Corn silage	2.9	3.0	11.0	8.5
Hay	1.8	0.6	1.4	0.3
Concentrate	5.1	8.8	9.2	9.1

PB: pasture-based farms; GR-TMR: grass silage-based total mixed ration; C-TMR: corn silage-based total mixed ration; GR/C-TMR: grass/corn silage-based total mixed ration.

**Table 2 animals-13-00574-t002:** Myristic (C14:0), palmitic (C16:0), stearic (C18:0), oleic (C18:1), saturated (SFA), monounsaturated (MUFA), polyunsaturated (PUFA) and short chain fatty acid (SCFA) (g/100 g fatty acids) quartile distributions in milk samples from dairy cows in Galicia (NW Spain).

	25	50	75
C14:0	9.7	11.1	12.2
C16:0	29.4	31.9	34.3
C18:0	8.7	9.6	10.8
C18:1	20.5	22.3	24.5
SFA	67.9	70.8	73.2
MUFA	23.5	25.5	28.0
PUFA	2.7	3.1	3.5
SCFA	9.8	10.4	11.0

**Table 3 animals-13-00574-t003:** Mean concentrations of myristic (C14:0), palmitic (C16:0) (and estimated CH_4_ emission), stearic (C18:0), oleic (C18:1), saturated (SFA), monounsaturated (MUFA), polyunsaturated (PUFA) and short chain fatty acid (SCFA) (g/100 g fatty acids) (dependent variables) in milk samples from dairy cows in Galicia (NW Spain) by type of ration, calving number, season and lactation phase (explanatory variables). The number of records for each category of the explanatory variables is also provided (n).

	C14:0	C16:0	CH_4_ estimation	C18:0	C18:1	SFA	MUFA	PUFA	SCFA
Type of ration									
Pasture (n = 3174)	10.3	30.9	442.9	10.5	24.1	69.4	27.5	3.5	10.6
Grass silage (n = 1630)	10.9	31.0	444.6	10.0	23.2	70.3	26.8	3.3	10.7
Corn silage (n = 2668)	11.5	33.4	484.7	9.9	22.5	71.9	26.0	2.9	10.6
Grass/corn silage (n = 2626)	11.4	32.7	473.1	9.8	22.4	71.5	25.8	3.0	10.7
Calving number									
First (n = 3089)	11.1	32.2	463.9	10.1	23.6	70.0	27.2	3.2	10.4
Second (n = 2651)	11.3	32.4	467.9	9.8	22.7	71.3	26.2	3.1	10.6
Third or higher (n = 4358)	10.8	31.8	457.3	10.1	22.9	70.9	26.3	3.2	10.7
Season									
Spring (n = 2774)	10.8	31.6	453.3	10.2	23.3	70.6	26.8	3.4	10.8
Summer (n = 1831)	11.0	32.2	464.8	10.2	23.5	70.6	26.7	3.2	10.1
Autumn (n = 2689)	11.2	32.3	465.7	9.8	22.9	70.8	26.3	3.0	10.6
Winter (n = 2804)	11.0	32.3	465.5	10.0	22.7	70.9	26.4	3.2	10.7
Lactation phase									
Fresh BHB < 0.10 mM/L (n = 746)	8.2	27.9	391.4	12.7	26.1	67.2	29.5	3.5	10.6
Fresh BHB ≥ 0.10mM/L (n = 168)	6.7	26.3	364.7	13.8	28.8	63.7	32.8	3.2	8.9
Peak (n = 2146)	10.3	31.9	458.8	10.7	23.3	70.4	26.6	3.2	10.8
Mid (n = 4180)	11.6	33.2	480.8	9.5	22.4	71.8	25.7	3.1	10.7
Late (n = 2858)	11.6	32.0	461.4	9.4	22.8	70.7	26.5	3.2	10.4

**Table 4 animals-13-00574-t004:** Results of mixed effects ordinal regression models for the effect of type of ration, calving number, season, lactation phase and energy corrected milk (explanatory variables) on mean amounts of myristic (C14:0), palmitic (C16:0), stearic (C18:0), oleic (C18:1), saturated (SFA), monounsaturated (MUFA), polyunsaturated (PUFA) and short chain fatty acid (SCFA) (g/100 g fatty acids) (dependent variables) in milk samples from dairy cows in Galicia (NW Spain). The regression coefficients—provided with their 95% confident intervals (CI)—indicate the likelihood of being in a higher category of the dependent variable when each category of explanatory variables is compared with the reference one (base). Random-effects cluster variance is provided (along with standard deviation (SD)).

	C14:0	C16:0	C18:0	C18:1	SFA	MUFA	PUFA	SCFA
	Variance (SD)	Variance (SD)	Variance (SD)	Variance (SD)	Variance (SD)	Variance (SD)	Variance (SD)	Variance (SD)
Herd	0.90 (0.95)	0.60 (0.77)	0.68 (0.83)	1.06 (1.03)	1.20 (1.09)	1.18 (1.08)	0.92 (0.96)	0.65 (0.80)
Herd:animal	1.45 (1.20)	1.40 (1.18)	1.34 (1.16)	1.31 (1.15)	1.65 (1.28)	1.69 (1.30)	1.37 (1.17)	0.86 (0.93)
	Coefficients (CI)	Coefficients (CI)	Coefficients (CI)	Coefficients (CI)	Coefficients (CI)	Coefficients (CI)	Coefficients (CI)	Coefficients (CI)
Feeding ^†^								
Grass silage	1.45 (0.50—4.16)	1.26 (0.52–2.99)	0.56 (0.22–1.41)		1.42 (0.42–4.75)		0.76 (0.26–2.19)	
Corn silage	3.73 ** (1.30–10.65)	5.54 ** (2.33–13.12)	0.40* (0.16–1.00)		4.76 ** (1.43–15.78)		0.16 ** (0.05–0.45)	
Grass/Corn silage	4.02 ** (1.40–11.46)	3.61 ** (1.52–8.52)	0.34 ** (0.13–0.84)		3.66 ** (1.10–12.13)		0.16 ** (0.05–0.47)	
Calving number ^‡^								
2nd	0.99 (0.83–1.17)	1.46 ** (1.23–1.71)	0.87 (0.74–1.02)	0.66 ** (0.55–0.77)	1.81 ** (1.52–2.14)	0.68 ** (0.57–0.81)	0.74 ** (0.62–0.87)	1.30 ** (1.10–1.52)
3rd or higher	0.57 ** (0.47–0.68)	1.04 (0.87–1.23)	1.21 ** (1.02–1.44)	0.76 ** (0.63–0.89)	1.68 ** (1.40–2.02)	0.78 ** (0.65–0.94)	0.99 (0.84–1.18)	1.63 ** (1.39–1.91)
Season§								
Summer	2.20 ** (1.91–2.51)	1.92 ** (1.68–2.18)	1.12 (0.98–1.28)	0.97 (0.85–1.10)	1.45 ** (1.27–1.66)	0.83 ** (0.72–0.94)	0.59 ** (0.52–0.67)	0.37 ** (0.32–0.42)
Autumn	2.32 ** (2.05–2.61)	1.81 *** (1.61–2.03)	0.52 ** (0.46–0.58)	0.75 ** (0.67–0.84)	1.50 ** (1.33–1.69)	0.69 ** (0.61–0.77)	0.24 ** (0.21–0.27)	0.97 (0.85–1.09)
Winter	1.36 ** (1.21–1.52)	1.36 ** (1.22–1.51)	0.53 ** (0.47–0.59)	0.52 ** (0.46–0.57)	1.39 ** (1.25–1.65)	0.58 ** (0.52–0.65)	0.40 ** (0.35–0.44)	0.78 ** (0.69–0.88)
Lactation phase ^¶^								
Fresh (BHB ≥ 0.10 mM/L)	0.27 ** (0.13–0.55)	0.46 ** (0.29–0.73)	4.40 ** (1.61–11.97)	4.84 ** (2.92–8.00)	0.24 ** (0.14–0.39)	5.72 ** (3.42–9.55)	0.32 ** (0.21–0.46)	0.06 ** (0.03–0.10)
Peak	13.90 ** (11.01–17.5)	9.15 ** (7.52–11.12)	0.04 ** (0.03–0.05)	0.14 ** 0.11–0.16	6.94 ** 5.70–8.45	0.15 ** 0.12–0.18	0.30 ** 0.25–0.36	1.73 ** 1.41–2.11
Mid	116.56 ** (91.85–147.65)	18.88 ** (15.59–22.86)	0.01 ** (0.00–0.07)	0.05 ** (0.04–0.06)	16.23 ** (13.38–19.68)	0.06 ** (0.05–0.07)	0.19 ** (0.15–0.22)	1.48 ** (1.22–1.78)
Late	126.28 ** (98.58–161.08)	9.23 ** (7.57–11.25)	0.01 ** (0.00–0.05)	0.07 ** (0.06–0.09)	9.17 ** (7.50–11.21)	0.11 ** (0.08–0.13)	0.23 ** (0.19–0.27)	1.03 (0.84–1.25)
Energy corrected milk yield (Kg)	1.05 ** (1.04–1.05)	1.00 (0.99–1.00)	0.96 ** (0.95–0.97)	0.95 ** (0.93–0.95)	1.03 ** (1.02–1.04)	0.94 ** (0.93–0.95)	0.99 ** (0.98–0.99)	1.05 ** (1.04–1.06)

^†^ Pasture is the base; ^‡^ 1st is the base; ^§^ Spring is the base; ^¶^ Fresh (BHB < 0.10 mM/L) is the base; * *p ≤* 0.05; ** *p* < 0.01; *** *p* < 0.001.

## Data Availability

Data available on request from the authors.

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
