# Peer review of "Factors Affecting Fatty Acid Composition of Holstein Cow’s Milk"

_animals, 2023, doi:10.3390/ani13040574_

Round 1

Reviewer 1 Report

General remarks:

The topic of this manuscript is interesting for Animals’ readers, but I find many problematic parts in the text.

Details

line 15: these parameters were not found in aims (lines 12-13)!

line 42: why estimated? Please rewrite this sentence, e.g. Based on previous results, mean FAs are..!

lines 44-45: other reports show other results!

lines 52-53: appr. 50-50% of C16:0 is originated from de novo and diet!

lines 76-79: aims are differ compared to earlier aims! But missing from these the health parameters, etc.!

2.1. subsection: please add information about the season!

lines 112-115: more information is needed about the feeding regime! 

line 120-121: how accurate is the FA analysis by Milkoscan? Some NIR devices are suitable for FA determination. How calibrated the device for FA? Maybe FA data only has an estimated value!

Table 1: in the text contains 4 groups, but in Table find only 3!

line 149: Appendix A – please give it as a separate file!

Table 2: P values? What does season mean Season? Month of calving?

Table 3: This Table is incomprehensible!

line 201: “…some FA and their variations…” What does it mean: FA and variations? Please clarify it!

line 227: ryegrass: may not be investigated in this study!

lines 232-235: no data diet FA composition!

lines 293-300: compare to introduction lines 55-60!

Conclusion: 1st sentence is not necessary, other parts of this section must be rewrite according to results! Please write an acceptable conclusion section!

Author Response

REVIEWER 1

General remarks:

The topic of this manuscript is interesting for Animals’ readers, but I find many problematic parts in the text.

Details

  1. line 15: these parameters were not found in aims (lines 12-13)!

We see the point and ms have been redrafted for a better comprehension.

  1. line 42: why estimated? Please rewrite this sentence, e.g. Based on previous results, mean FAs are..!

Thank you for your comment, the referee was right and the manuscript has been amended in this sense.

  1. lines 44-45: other reports show other results!

We are grateful for the comment. An explanation for better comprehension has been added.

  1. lines 52-53: appr. 50-50% of C16:0 is originated from de novo and diet!

The referee is right. The ms has been corrected.

  1. lines 76-79: aims are differ compared to earlier aims! But missing from these the health parameters, etc.!

We appreciate the comment. The reason for introducing that comment was showing that fatty acid profile was important because of its implication in other parameters previously studied as health, organoleptic properties, etc. Due to this importance, it is useful to study their variation according to type of ration, parity, lactation phase and season. For a better understanding we have clarified the ms.

  1. 1. subsection: please add information about the season!

Thank you for your appreciation. Information about season categorization was included in the ms.

  1. lines 112-115: more information is needed about the feeding regime! 

We understand the referee’s concern. We have included a characterization table (Table 1) were feeding regime is described. Furthermore, we would like to emphasize that the purpose of this study was to value the data recorded by Dairy Herd Improvement Program (DHIP), particularly fatty acid composition according to the type of forage used for feeding animals.

  1. line 120-121: how accurate is the FA analysis by Milkoscan? Some NIR devices are suitable for FA determination. How calibrated the device for FA? Maybe FA data only has an estimated value!

We understand the referee’s concern about Milkoscan accuracy, however, the data used for this study are originated in Dairy Herd Improvement Program (DHIP). DHIP obtained milk samples and had delivered them to the official laboratory (LIGAL-Laboratorio Interprofesional Galego de Análise do Leite). LIGAL is declared official laboratory according to Decreto 120/2011 and functions following official criteria UNE-EN ISO/IEC 17025, with nº 99/LE267.

We have clarified the officiality of the analysis in the ms, for a better understanding.

  1. Table 1: in the text contains 4 groups, but in Table find only 3!

We understand the referee’s concern. Even that FA concentrations have been divided in 4 categories, there are 3 quartiles of distribution that function as cut points helping to stablish the 4 categories. In this sense, the ms has been clarified for a better comprehension.

  1. line 149: Appendix A – please give it as a separate file!

According to the authors’ guidelines it is not possible to separate the appendix in a new file. As stated in authors’ guidelines ‘Appendixes must be cited in the main text.’.

  1. Table 2: P values? What does season mean Season? Month of calving?

We are grateful for your comment. Table 2 reflects FA means depending on type of ration, calving number, season and lactation phase. All the traits studied were significant in the univariate analysis (comparisons within each of the variables studied). To facilitate reading, this is mentioned in a generic way and now it is added that a p<0.05 was considered significant instead of mentioning all the p values (9 traits for 4 variables analyzed implicitly providing 36 p values). If the reviewer considers it important to contribute, all of them could be included in a subsequent review. Information about season meaning was included in material and methods.

  1. Table 3: This Table is incomprehensible!

To the explanation provided about ordinal regression in the statistical analysis part, this sentence "Thereby, the regression coefficients indicate the likelihood of being in a higher ranking, when the explanatory variables change" has been added, as well as "For random factors, the cluster variance was provided in terms of an intraclass correlation" when it is mentioned that herd and cow were included as random factors, to make adjustments within herd and animal cluster effects. Likewise, along the lines mentioned in the statistical analysis, the header of the table has been expanded: "Results of mixed effects ordinal regression models for the effect of type of ration, calving number, season, lactation phase, and energy corrected milk (explanatory variables) on mean amounts of myristic (C14:0), palmitic (C16:0), stearic (C18:0), oleic (C18:1), saturated (SFA), monounsaturated (MUFA), polyunsaturated (PUFA) and short chain fatty acid (SCFA) (g/100 g fatty acids) (dependent variables) in milk samples from dairy cows in Galicia (NW Spain).The regression coefficients -along with their 95% confident intervals (CI)- indicate the likelihood of being in a higher category of the dependent variable, when each category of explanatory variables is compared with the reference one (base). Random-effects cluster variance is provided (along with standard deviation (SD))".

  1. line 201: “…some FA and their variations…” What does it mean: FA and variations? Please clarify it!

The author is true, the writing could be better explained, so the ms has been corrected for a better comprehension.

  1. line 227: ryegrass: may not be investigated in this study!

We are grateful for the comment. We have mentioned ryegrass as we are referencing the use of this type of forage from the original paper where it was used. Nevertheless, we have redrafted the ms.

  1. lines 232-235: no data diet FA composition!

We apologize for the writing. We would like to explain that we did not find such marked differences in cows’ milk after consuming fresh or conserved forage. So, we have reviewed and corrected the ms for a better comprehension.

  1. lines 293-300: compare to introduction lines 55-60!

We see the point. We have redrafted the ms for a better understanding.

  1. Conclusion: 1stsentence is not necessary, other parts of this section must be rewrite according to results! Please write an acceptable conclusion section!

We are grateful for the comment and we have remade the conclusions section.

Reviewer 2 Report

The procedure to select randomly the dairy farms is not well explained. 

Was it simple, sistematic or stratified? How the procedure was carried out?

Author Response

Comments and Suggestions for Authors

  1. The procedure to select randomly the dairy farms is not well explained. 

We appreciate the comment. This study is part of a major research project (FEADER 2016/59B). Farms belonging to this study were selected by simple random sampling as explained in material and methods section: ‘The data used in the study were obtained from 1,557 cows (all of which are Holstein breed) belonging to 25 dairy farms selected by simple random sampling among those included in the DHIP’.

  1. Was it simple, sistematic or stratified? How the procedure was carried out?

Simple random sampling was performed and it is indicated now in the text.

Reviewer 3 Report

Although the working title caught my attention quite a bit, I think it is quite incompatible with the content.

The study does not contain a fully planned trial pattern. It is a study that will only affect milk quality, and most importantly, as a result of bringing together known facts. I think it was a waste of time to analyze so many milk samples to determine this. Perhaps better results could have been obtained if a meta-analysis had been done.

In the study, information about the farms sampled should be added to the material and method. For example; such as herd size, animal breed, breeding.

Apart from this, for the determination of milk fatty acid, samples should be collected in a second tube that does not contain antibiotics and stored in the freezer until analysis.

It would be better to show the methane estimation result in a separate column in the table.

As a result of all these explanations, it is appropriate for the article to be evaluated as a "brief report".

Author Response

REVIEWER 3

Comments and Suggestions for Authors

  1. Although the working title caught my attention quite a bit, I think it is quite incompatible with the content.

We think the title is compatible with the content of the study developed. The title proposed is ‘Factors affecting fatty acid composition of Holstein cow’s milk’. In the study we are analysing how type of ration, parity, lactation phase and season condition  fatty acid composition of Holstein dairy cows, so that are the four factors that we are studying. However, we are opened to listen referee’s suggestions.

  1. The study does not contain a fully planned trial pattern. It is a study that will only affect milk quality, and most importantly, as a result of bringing together known facts. I think it was a waste of time to analyze so many milk samples to determine this. Perhaps better results could have been obtained if a meta-analysis had been done.

Thank you for your comment, we see the point of view of the referee however data analyzed in this study are originated in Dairy Herd Improvement Program (DHIP)so we do not have to perform this analysis just for the study. In fact, there are not field farm condition papers that analyses a large sample size, so that is the purpose and the relevance of the study where a large amount of data are analyzed in field farm conditions.

We are aware of the existence of similar studies mostly developed in research conditions that characterize different factors affecting fatty acid profile. However, this is the first analysis of fatty acid profile developed in our region, and it is important to know how these 4 factors affect fatty acid profile in our specific field conditions. In addition, these date could help to compare our results with other papers published internationally.

  1. In the study, information about the farms sampled should be added to the material and method. For example; such as herd size, animal breed, breeding.

Thank you for this comment. Herd size has been included.

In the case of animal breed it was already reflected that cows are Holstein breed in material and methods section ‘The data used in the study were obtained from 1,557 cows (all of which are Holstein breed)...’.

In the case of breeding system, farms included in this study belong to DHIP, indeed, animals belonging to those herds are part of a controlled breeding program.

  1. Apart from this, for the determination of milk fatty acid, samples should be collected in a second tube that does not contain antibiotics and stored in the freezer until analysis.

We understand the concern of the referee about the sample collection, however, the data used for this study are originated in Dairy Herd Improvement Program (DHIP). DHIP obtained milk samples and had delivered them to an official laboratory (Ligal-Laboratorio Interprofesional Galego de Análise do Leite). Ligal laboratory functions following official criteria UNE-EN ISO/IEC 17025, with nº 99/LE267.

Although, data used in these study are collected and analyzed following the compulsory processes officially determined, we see the point of the referee, it is true that a more sophisticated analysis could be performed. However, we would like to emphasize, once again, that it is a study developed in field farms conditions and we consider that the type of analyses performed does not affect the possibility of accurately studying factors influencing fatty acid composition.

  1. It would be better to show the methane estimation result in a separate column in the table.

We accept the point. The ms has been corrected. 

  1. As a result of all these explanations, it is appropriate for the article to be evaluated as a "brief report".

This type of study, due to the magnitude of data, is difficult to make more concise. We think that it is not feasible to adapt it to a brief report. In fact, as it is stated in guidelines a brief report contains about 2,500 words and two figures and/or a table. In the case of our study, it contained three tables and one of the referees has asked as for new data so we needed to add new information in another table. 

Reviewer 4 Report

The manuscript deals with the influence of various factors on the fatty acid profile in the milk of Holstein cows. The manuscript requires many corrections, especially in the material and methods chapter. The authors should also improve the conclusions in this paper.

Line 21 : Correct "animals" to "cow's milk".  The authors need to correct this throughout the manuscript, after all they were studying milk not animals.

Line 88: The authors only gave the total number of cows. They should complete the information with the number of cows in each group or give the number of milk samples in each group e.g. spring (n=?), autumn (n=?).....

Also for other factors such as type of ration, calving number and lactation phase.

The authors should add information on cow nutrition and rations in the chapter on 'metering'.

The subsection "test methods" should be separated and describe which method was used to determine fatty acids, milk chemistry and β-hydroxybutyrate (BHB) concentration.

Lines 169-171. The authors write: ” This section may be divided by subheadings. It should provide a concise and precisedescription of the experimental results, their interpretation, as well as the experimental  conclusions that can be drawn.” I think so too. The authors need to complete it.

Furthermore, conclusions do not follow from the results of the study. The authors should relate their conclusions to their own research and not to the abstract results of an organoleptic evaluation.

Author Response

REVIEWER 4

Comments and Suggestions for Authors

  1. The manuscript deals with the influence of various factors on the fatty acid profile in the milk of Holstein cows. The manuscript requires many corrections, especially in the material and methods chapter. The authors should also improve the conclusions in this paper.

We accept the point. Material and methods have been improved and conclusions have been redrafted.

  1. Line 21 : Correct "animals" to "cow's milk".  The authors need to correct this throughout the manuscript, after all they were studying milk not animals.

We appreciate the comment. Ms has been amended in this sense.

  1. Line 88: The authors only gave the total number of cows. They should complete the information with the number of cows in each group or give the number of milk samples in each group e.g. spring (n=?), autumn (n=?).....

We accept the point. Number of samples have been included in table 3.

  1. Also for other factors such as type of ration, calving number and lactation phase.

We indicated in the material and methods section that “the study included data from 10,098 test-days (obtained from the 1,557 cows). Each test day included:....” number of milk samples per explanatory variable category were now included in table 3.

  1. The authors should add information on cow nutrition and rations in the chapter on 'metering'.

Despite of knowing that this detailed information would be interesting, we do not have deeply access to these data for each farm. Even though, we have data of daily ration composition that have been included in Table 1. The objective of this study was valorize data collected by Dairy Herd Improvement Program (DHIP) classifying data according to forage type. This type of classification makes it more affordable for its use in field farm conditions.

  1. The subsection "test methods" should be separated and describe which method was used to determine fatty acids, milk chemistry and β-hydroxybutyrate (BHB). concentration.

The data used for this study are originated in Dairy Herd Improvement Program (DHIP). DHIP obtained milk samples and had delivered them to an official laboratory (LIGAL-Laboratorio Interprofesional Galego de Análise do Leite). LIGAL laboratory functions following official criteria UNE-EN ISO/IEC 17025, with nº 99/LE267.

  1. Lines 169-171. The authors write: ” This section may be divided by subheadings. It should provide a concise and precise description of the experimental results, their interpretation, as well as the experimental  conclusions that can be drawn.” I think so too. The authors need to complete it.

Although it was a proposal that we considered, in the end, in this section it was decided to strictly reflect results, leaving results assessments for the discussion and conclusions section, trying to be concise and avoiding being repetitive and unnecessarily lengthening the paper. If deemed appropriate it could be addressed in a subsequent review.

  1. Furthermore, conclusions do not follow from the results of the study. The authors should relate their Rconclusions to their own research and not to the abstract results of an organoleptic evaluation.

We are grateful for the comment, and we have remade the conclusions section.

Round 2

Reviewer 1 Report

All sections of this manuscript were improved by authors, so I recommend this manuscript for publishing in the Animals journal.

I recommend editing the Tables (3rd and 4th)! The table should be placed on a new page!

And I also recommend removing Appendix A from the text, please attach as a separate file!

Reviewer 4 Report

The authors have corrected most of my comments. They provided comprehensive answers to the remaining comments. I have no further comments.